# The Effects of Pharmacological Treatment of Nightmares: A Systematic Literature Review and Meta-Analysis of Placebo-Controlled, Randomized Clinical Trials

**DOI:** 10.3390/ijerph20010777

**Published:** 2022-12-31

**Authors:** Mathilda Skeie-Larsen, Rebekka Stave, Janne Grønli, Bjørn Bjorvatn, Ane Wilhelmsen-Langeland, Amin Zandi, Ståle Pallesen

**Affiliations:** 1Department of Psychosocial Science, University of Bergen, 5007 Bergen, Norway; 2Department of Biological and Medial Psychology, University of Bergen, 5007 Bergen, Norway; 3Norwegian Competence Center for Sleep Disorders, Haukeland University Hospital, 5021 Bergen, Norway; 4Department of Global Public Health and Primary Care, University of Bergen, 5007 Bergen, Norway; 5Bjørgvin District Psychiatric Center, Division of Psychiatry, Haukeland University Hospital, 5021 Bergen, Norway; 6Faculty of Psychology and Educational Sciences, University of Tehran, Tehran 1417935840, Iran

**Keywords:** nightmare, pharmacological intervention, RCT, meta-analysis, PTSD

## Abstract

Nightmares are highly prevalent and distressing for the sufferer, which underlines the need for well-documented treatments. A comprehensive literature review and meta-analysis of the effects of different pharmacological placebo-controlled randomized clinical trials, covering the period up to 1 December 2022, was performed. Searches were conducted in PubMed, Embase, Web of Science, PsychInfo, Cinahl, and Google Scholar, resulting in the identification of 1762 articles, of which 14 met the inclusion criteria: pharmacological intervention of nightmares, based on a placebo-controlled randomized trial published in a European language, reporting outcomes either/or in terms of nightmare frequency, nightmare distress, or nightmare intensity, and reporting sufficient information enabling calculation of effect sizes. Most studies involved the effect of the α_1_-adrenergic antagonist prazosin in samples of veterans or soldiers suffering from posttraumatic stress disorder. Other medications used were hydroxyzine, clonazepam, cyproheptadine, nabilone, and doxazosin. The vast majority of studies were conducted in the USA. The studies comprised a total of 830 participants. The Clinician-Administered PTSD Scale was the most frequently used outcome measure. The results showed an overall effect size of Hedges’ *g* = 0.50 (0.42 after adjustment for publication bias). The synthetic cannabinoid nabilone (one study) showed the highest effect size (*g* = 1.86), followed by the histamine H_1_-antagonist hydroxyzine (one study), and prazosin (10 studies), with effect sizes of *g* = 1.17 and *g* = 0.54, respectively. Findings and limitations are discussed, and recommendations for future studies are provided.

## 1. Introduction

Frequent and vivid recall of dreams occurs in approximately 80% of cases where individuals are awakened from rapid eye movement (REM) sleep [1]. Dreams can also occur during non-REM (NREM) sleep. However, dreams occurring during REM sleep are often more complex, vivid, bizarre, last longer, and are associated with content that revolves around visually perceptual and emotional memories [2]. Some dreams become nightmares, defined as vivid dreams with strong and negative emotional content. They often lead to such high levels of activation that the dreamer awakens [3]. Nightmares occur during REM sleep, but posttraumatic nightmares can also occur during NREM sleep. After waking from nightmares, the dreamer rapidly becomes alert and oriented and can easily remember and reproduce the content, which in most cases involves threats to life, security, or self-esteem [4].

The most common emotions associated with nightmares are fear-related, but other less common emotions such as anger and disgust have also been reported [5]. Several scholars distinguish between idiopathic and posttraumatic stress disorder (PTSD)-related nightmares, and physiological differences between those suffering from the two forms of nightmares have been described [6]. Idiopathic nightmares are often illogical and bizarre [6]. PTSD-related nightmares are often considered a more severe form [5]. PTSD is a psychiatric disorder that may develop after a traumatic experience, and flashbacks of the event alongside, or in the form of, nightmares, are two core symptoms.

In the general adult population, the prevalence of nightmares is relatively high, where 2–6% of adults report having weekly nightmares, while 35–45% report having nightmares at least once a month [7,8,9,10,11,12]. According to most studies, nightmares also appear to be age-related, with the highest prevalence among children and adolescents [13,14]. The prevalence of nightmares is elevated in populations with mental disorders, especially in anxiety, mood, and psychotic disorders [15,16,17,18]. Furthermore, nightmares are more common in populations with a high probability of being exposed to traumatic events, such as war veterans [11]. The healthcare and societal costs of nightmares seem largely unknown, but the costs of PTSD, where nightmare is a common symptom, amounted to USD 232.2 billion for 2018 in the USA alone [19]. Nightmares are reported more frequently after being exposed to traumatic life events (e.g., physical and sexual abuse, witnessing or victimization of accidents, domestic violence, war experience, and terrorism), and during periods of high levels of stress in general [10,20,21,22,23,24]. The association between nightmares and stress-related symptoms and conditions is in line with the notion that nightmares can be linked to dream functions involving regulation of fear and other emotions [4,5,25]. Women report more nightmares than men, although this difference is largest among young adults [5,11,26]. Several drugs such as amphetamines, anticholinergics, benzodiazepines, dopamine agonists, ethanol, hallucinogenics, and tricyclic antidepressants may induce nightmare, either following intake or due to withdrawal effects [27].

Some regard nightmares as non-functional, whereas other scholars view nightmares as a mechanism for emotion regulation. The functional theories vary when it comes to the specific mechanism involved, including among others, stress mastery, desomatization of affect, fear memory extinction, and emotional contextualization [28]. Regardless of its functionality, frequent nightmares significantly deteriorate quality of life and are also associated with suicidal thoughts, suicide attempts, and suicide [28]. Nightmares may also lead to avoidance of going to sleep in fear of further nightmares, which may lead to sleep deprivation, which in turn exacerbates nightmares [29]. Nightmares may also predispose for insomnia [30]. This emphasizes the importance of efficacious treatments.

Currently, several psychological treatments have been developed, of which Imagery Rehearsal Therapy (IRT) is recommended for both PTSD-related and idiopathic nightmares [31]. IRT involves a positive rewriting of the negative content in nightmares. The rewriting should be visualized in the awake state for 10–20 min every day [32,33], and it is assumed that the positive images gradually become more dominant and accessible when dreaming. Other psychological treatments that are empirically supported for idiopathic nightmares are systematic desensitization, self-exposure therapy, lucid dreaming, and multicomponent therapies [31]. The multicomponent approach has received increased attention during the last years, with exposure, relaxation, and rescripting therapy (ERRT) as a prominent example, where IRT is supplemented with an exposure component [34].

Regarding pharmacological treatment, studies have largely focused on their effect on PTSD-related nightmares. Several pharmaceuticals have been studied, in particular prazosin, which is an α_1_-adrenergic antagonist assumed to inhibit fear responses by, among others, blocking α_1_-adrenergic receptors in the brain [35]. Prazosin was marketed as an antihypertensive drug but was by chance found to reduce PTSD-related nightmares [36]. That finding led to several placebo-controlled, randomized clinical trials comparing the efficacy of prazosin against placebo [37,38,39,40,41,42,43,44,45,46]. Several recent meta-analyses show that prazosin is associated with statistically significant improvement in PTSD-related nightmares [47,48,49,50], and Morgenthaler et al. [31] highlight prazosin as one of the preferred drugs to prescribe for PTSD-related nightmares.

One of the studies examining prazosin [37] also examined the effect of hydroxyzine, a histamine-H_1_-antagonist that has been shown to influence sleep and stress in patients with PTSD [51,52]. Exactly how hydroxyzine influences sleep is unknown, but it is known that hydroxyzine, in addition to acting as an H_1_-antagonist, also has some affinity for the serotonin 5H_2A_-, the dopamine D_2_-, and the α1-adrenergic receptors [53,54,55]. Regarding other pharmacological agents studied regarding the effect on nightmares, one finds cyproheptadine, a 5-HT_2_ antagonist, which has been used to increase the amount of NREM sleep [56] and improve sleep in general [57].

Clonazepam, a high-potency drug that binds to benzodiazepine α-receptors and potentiates GABA-ergic inhibition has also been investigated. The drug is used to treat epileptic seizures [58], but an open study [59] showed that clonazepam had a good effect on PTSD-related nightmares. Treatment studies have further examined the effects of nabilone [60] and doxazosin [61]. Nabilone is a synthetic cannabinoid that activates cannabinoid-_1_ receptors (CB_1_) and has long been approved for the treatment of chemotherapy-induced nausea and vomiting [62]. However, it has also reduced PTSD-related nightmares [63]. Doxazosin, which is also prescribed against nightmares, is, like prazosin, an α_1-_antagonist [64].

When it comes to pharmacological interventions, there is currently no firm basis for making clear recommendations, as studies based on placebo-controlled designs are relatively few. Under these circumstances, it is essential to produce summaries of effects across the existing studies. As far as we know, it has been almost 10 years since the last time this was done [65]. Against this backdrop, we conducted a systematic literature review and meta-analysis to investigate how different pharmacological interventions affect nightmare symptomatology in studies where placebo comprised the control condition. Specifically, we aimed to estimate an overall effect size based on a random effects model, as well as effect sizes (Hedges’ *g*) for the different drugs separately, as well as estimating heterogeneity and trim-and-fill results as adjustment for potential publication bias. In order to evaluate the robustness of the findings, we used the Knapp–Hartung adjustment [65], and also ran a separate meta-analysis only including studies with low risk of bias.

Such a literature review should guide clinicians by pointing out the best documented and most effective pharmacological approaches, and pointing out future research needs in the field.

## 2. Materials and Methods

### 2.1. Registration and Reporting

The meta-analysis was pre-registered in the International Prospective Register of Systematic Reviews (PROSPERO: CRD42022314503). The updated PRISMA guidelines [66] were followed to ensure transparent reporting of the procedures.

### 2.2. Literature Search

A literature search was conducted in the databases PubMed, Embase, Web of Science, PsychInfo, Cinahl, and Google Scholar (200 first hits sorted by relevance). The searches covered the period up to December 1 of 2022. The initial search strategy was based on a combination of relevant keywords: nightmare * AND («drug therapist *» OR «pharmacological treatment») AND («randomized controlled trial * OR «randomized controlled trial *» OR RCT OR «controlled clinical trial *» OR randomize * OR randomize *). To extend the search to include more potentially relevant studies, a supplementary search was also carried out and was based on the following combinations of keywords: nightmare * AND («randomized controlled trial *» OR «randomized controlled trial *» OR randomize * OR randomize * OR placebo) AND (prazosin OR nabilone OR zolpidem OR cannabinoid OR clonazepam OR levomepromazine OR cyproheptadine OR hydroxyzine OR hydroxytryptophan). The reference lists of relevant publications and previous literature reviews on the topic were also inspected. The search and identification of relevant studies was conducted independently by two (MSL and RS) of the authors. Percentage agreement for inclusion/exclusion was calculated. In cases of disagreement, this was resolved by discussions.

### 2.3. Inclusion Criteria

The meta-analysis included randomized clinical trials (RCTs) that: (1) were placebo-controlled, (2) involved a pharmacological intervention targeting nightmares, (3) were published in a European language, and (4) reported outcomes related to nightmares in at least one of three following measures: (i) distress, (ii) intensity, (iii) or frequency, and (5) reported sufficient information to calculate an effect size. The participants were not required to have a formal diagnosis of nightmare disorders. In cases of missing data, we contacted the authors via e-mail requesting the missing information.

### 2.4. Coding of the Studies

Prior to the meta-analysis, all included studies were coded for raw data. To ensure quality and to avoid errors, all coding was conducted by two authors (MSL and RS) independently of each other. The coding was carried out using a prepared codebook based on the PICO variables: population, intervention, comparison, and outcome. Specifically, the studies were coded for (i) participant characteristics (number, age, gender, and diagnosis/problem), (ii) intervention characteristics (type of pharmaceutical, dosage, and duration of intervention), (iii) control group (type), and (iv) outcome variables (outcome measure, follow-up according to baseline, percentage dropout and method for adjustment of dropout). Study characteristics such as authors, year of publication, type of manuscript, study design, country, and the continent of publication were also coded. Finally, it was coded whether included studies reported an a priori power analysis. In cases of missing data, we contacted the authors via e-mail requesting the missing information.

### 2.5. Risk of Bias in Included Studies

To evaluate the risk of bias in the included studies, the Cochrane Risk of Bias 2 tool (RoB 2) was used [67]. RoB 2 contains five domains for assessing the risk of bias: (1) Biases that occur in the randomization process, (2) biases due to deviations from intended interventions, (3) biases due to lack of outcome data, (4) biases in the measurement of outcomes, and (5) biases in the selection process of reported results. For each domain in RoB 2, studies are considered to have either «low risk», «some risk», or «high risk». The included studies were evaluated by two of the authors (MSL and RS) independently. The percentage agreement between the two raters was calculated for each domain. Disagreements were followed by discussions to reach consensus.

### 2.6. Statistical Analysis

The statistical analysis was performed by Comprehensive Meta-Analysis, version 3.0. The analysis aimed to calculate how strong an effect the pharmacological interventions had compared to placebo. Since few studies reported data from follow-up, it was decided to calculate effect sizes by comparing the intervention and placebo groups at the end of treatment. For one study [44], the effect size was calculated from an independent sample t-test. In the cases where a crossover design was used, the correlation coefficient between the two measurements was set to 0.50, in line with general recommendations [68].

Since most included studies reported only one outcome measure related to nightmares (discomfort, intensity, frequency), we decided to conduct one meta-analysis for nightmare symptomatology overall. In studies that reported more than one outcome, an average of relevant outcome measures was therefore calculated before the data were entered into the analysis program. In these cases, the correlation coefficient between the various outcome measures was set at 0.7. This approach provides a better estimate of the variance of the outcome measures than assuming a correlation coefficient of 1.0, which is the standard in most meta-analytic software [69].

For all studies, the direction of the effect size (Hedges’ *g* with associated 95% confidence interval) was calculated, so that positive effect sizes indicated larger symptom relief for the intervention group compared to placebo. Thus, negative effect sizes suggest that the placebo group fared better than the intervention group. The overall effect size was calculated based on a random-effects model where the procedure of DerSimonian and Liard (1986) was used. Heterogeneity between studies was calculated in terms of Cochran’s Q. I^2^ values were also calculated and indicate how much of the variance in the observed effects reflects real variance [69], where 0.25, 0.50, and 0.75 are regarded as small, medium, and large, respectively [70].

Publication bias was visually examined using a funnel plot. The funnel plot was supplemented by Duval and Tweedie’s [71] “trim and fill” procedure, which removes and imputes studies so that the funnel plot becomes symmetrical; it is thus used for calculating unbiased effects. To examine the stability of the findings, Orwin’s “Fail-safe N” [72] was calculated, assessing, in this case, how many studies with zero effect (*g* = 0.00) are needed to bring the overall effect size down to a trivial level (*g* = 0.20). In the analysis, medication was used as a moderator. In one study [37], two drugs were tested against placebo. To prevent the subjects in the placebo group from being weighted twice, which would result in an underestimation of the variance, this was, in line with current recommendations [73], corrected for by dividing the number of subjects in the placebo group in two. In order to investigate the robustness of the conclusion, the results were also calculated with the Knapp–Hartung adjustment [74]. In addition, we conducted a separate analysis, only including studies which were deemed to have low risk of bias.

## 3. Results

### 3.1. Literature Search

Through the literature search and the supplement search, a total of 1762 articles were identified. After manual removal of duplicates, a total of 1182 articles were reviewed. Based on title and abstract, 1161 articles were excluded with an inter-rater agreement of 98.4%. The remaining 21 articles were thoroughly reviewed in full text. Seven of them did not meet the inclusion criteria and were thus excluded. A flow chart of the literature search can be seen in Figure 1. A total of 14 articles met all inclusion criteria and were included in the meta-analysis.

### 3.2. Characteristics in Included Studies

Table 1 provides an overview of the characteristics of the included studies. Of the fourteen included studies, twelve were conducted in the United States [38,39,40,41,42,43,44,45,46,61,75,76,77], one in Canada [58], and one in Switzerland [35]. Five of the studies had a crossover design in which participants received both drug treatment and placebo [43,46,60,61,76]. The remaining nine studies had between-group designs [37,38,39,40,41,44,45,77]. In these, participants were randomized to receive either a pharmacological intervention or placebo, except in one study [38], in which 24 participants were randomized to a third condition (cognitive behavioral therapy). In total, the included studies comprised 830 participants. The studies with a between-group design included 783 participants, 398 participants in the pharmacological conditions and 361 participants in placebo groups (and 24 who received cognitive behavioral therapy) [76].

Of the 14 studies, participants in nine of them were soldiers or veterans with war-related PTSD [38,41,42,43,44,60,61,75,76], and participants in one study were veterans with comorbid war-related PTSD and alcoholism [40]. Two studies included patients with non-war PTSD [37,46], while the participants in one study suffered from comorbid non-war-related PTSD and alcoholism [45]. Finally, participants in one study were PTSD patients with suicidal ideation [39]. Thus, all participants in the 14 studies had PTSD-related nightmares.

In terms of the outcome measures, eight studies [40,41,43,44,46,60,61,75] measured nightmare symptomatology using the Clinician-Administered PTSD Scale (CAPS) [77]. Raskind and colleagues [42] used the Nightmare Frequency Questionnaire (NFQ) [78] in addition to CAPS. Two studies [37,76] used sleep- and nightmare-related questions from the Pittsburgh Sleep Quality Index (PSQI) [79] to measure nightmare symptomatology. Germain and colleagues [38] asked participants to keep a sleep diary where they recorded the total number of nightmares per week. McCall and colleagues [39] used the Disturbing Dreams and Nightmare Severity Index (DDNSI) [80], while Simpson and colleagues [45] used a question from the PTSD Checklist-Civilian Questionnaire version (PCL) [81] in order to assess nightmare symptomatology.

The mean age across the studies was 43.9 years. The authors of one of the included studies [76] did not state the age of the participants. In the included studies, 20% of the participants were women. Overall, the studies had an average dropout rate of 20%. Furthermore, only two [41,75] of the fourteen studies reported a priori power calculations.

### 3.3. Interventions

The 14 included studies comprised a total of 15 pharmacological interventions. Nine of the interventions were based on prazosin [38,39,40,41,42,43,44,45,46]. One study [37] included both a prazosin and hydroxyzine condition. The remaining four studies involved pharmacological interventions with clonazepam [75], cyproheptadine [76], nabilone [60], and doxazosin XL [61], respectively. In all studies, the dosage of the drugs was titrated to avoid acute onset side-effects. The mean duration of intervention was 6.5 weeks, varying from two to 12 weeks. See Table 1.

### 3.4. Risk of Bias in Included Studies

There was an overall agreement on 53 out of 70 parts evaluated for risk of bias by the two assessors, amounting to an agreement of 75.7%. Of the fourteen studies, nine exhibited risks of bias to varying degrees. The risk of bias analysis is depicted in Table 2.

### 3.5. Results at the End of Treatment

The results of the 15 pharmacological interventions after treatment (14 studies, N = 830) showed an overall effect size of *g* = 0.50 (95% CI = 0.14–0.87, *p* = 0.007). See Figure 2. It should be noted that there was large variation in the effect sizes between the different studies manifested by a Cochran’s Q of 83.1 (*df* = 14, *p* < 0.01), which indicates significant heterogeneity. The I^2^ suggested that 83.2% of the variation in effect sizes reflects real variation. Following the Knapp–Hartung adjustment [72], the 95% CI interval became wider (0.06–0.95, *p* = 0.029), but the overall effect size was still significant. When only including studies with low risk of bias (k = 5), the overall effect size became non-significant (*g* = 0.11, 95% CI = −0.44–0.67, *p* = 0.690).

The funnel plot (see Figure 3) indicated that publication bias was present. The studies with small samples seemed to provide more positive results than those with larger samples. The trim-and-fill procedure [68] revealed that one study should be imputed to make the funnel plot symmetrical. Adjusted calculations after the trim and fill procedure yielded an imputed overall effect size of 0.42 (95% CI = 0.05–0.79). Orwin’s Fail-safe N was equal to 19. To investigate whether there were differences in treatment effect between the different pharmaceuticals, a sub-group analysis was conducted. The results were significant (Q_bet_ = 27.45, *df* = 5, *p* < 0.01) and showed that nabilone had the largest effect, *g* = 1.86 (95% CI = 0.87–2.85); see Figure 3. Hydroxyzine had the second highest effect (*g* = 1.17 (95% CI = 0.54–1.80). Prazosin also had a positive effect with an overall effect size of *g* = 0.54 (95% CI = 0.10–0.99). Three pharmaceuticals had no effect: clonazepam (*g* = −0.11 (95% CI = −0.75–0.52), cyproheptadine (*g* = −0.35 (95% CI = −0.86–0.15), and doxazosin (*g* = –0.04 (95% CI = −0.65–0.58).

## 4. Discussion

In this systematic literature review and meta-analysis, the main findings showed a significant overall and positive effect size of pharmacological interventions for nightmares. Our findings thus differ from the meta-analysis of Augedal and colleagues [65], where the authors concluded that pharmacological interventions for nightmares did not have a significant effect. The discrepancy in results probably reflects that several studies have emerged since then. Recent studies have also examined effects of new types of pharmaceuticals, such as nabilone [60]. Despite the fact that we found an overall significant effect of pharmacological treatments of nightmares, it can be considered small to moderate (*g* = 0.42) following adjustment for publication bias.

The heterogeneity of the analysis was significant and indicated differences in treatment effects between the various drug interventions. A possible explanation may be that the included studies examined drugs with different mechanisms of action. Still, the heterogeneity may also reflect several differences between the studies in terms of sex, age, comorbidity, dosage, treatment duration, and outcome measures. The overall effect size can thus be somewhat misleading regarding the effect of the individual drugs. Therefore, a sub-group analysis, broken down by specific groups of drugs, was conducted. Nabilone was the drug with the largest effect (*g* = 1.86) and is included in current treatment recommendations [31]. It is still worth mentioning that the finding is based on a single RCT-study [60] and that the sample size was small. In addition, the study had some risk of bias. Still, the findings of Jetly and colleagues [60] are nevertheless promising, and it is essential to note that nabilone appears to be well tolerated [81]. Thus, further research on nabilone in the treatment of nightmares is encouraged.

The second most effective pharmaceutical was hydroxyzine (*g* = 1.17). As for nabilone, only one RCT examined this pharmaceutical [37]. That study, with a sample size of 34 participants in the hydroxyzine condition, showed that hydroxyzine was well tolerated and associated with few side-effects. On the other hand, the study had some limitations and was regarded to have a high risk of bias related to missing outcome data.

Prazosin is the drug that has been most studied and was, in the present meta-analysis, investigated across 10 trials, with an overall effect size of *g* = 0.54. This finding is in line with previous meta-analyses of prazosin’s effect on nightmares [47,48] and in line with “Best practice” guidelines based on an initiative from the American Academy of Sleep Medicine [29]. Notably, the effect of the individual prazosin studies in the analysis diverged. In this regard, the finding by McCall et al. [39] stood out, where prazosin had a negative effect on nightmare symptomatology (*g* = −1.81). McCall et al. [39] point out that this divergent finding may result from the sample they included, which was small and comprised highly suicidal patients taking a complex composition of psychotropic drugs. Raskind et al. [40] reported no effect of prazosin (*g* = 0.00). That study was the largest of the ones included. Following the finding by Raskind et al. [41], the treatment recommendation regarding prazosin was downgraded; still, Morgenthaler et al. [31] argue that prazosin is a drug that seems well tolerated, without special side-effects, and thus is relevant in treating PTSD-related nightmares. Taken together, the divergence in the findings regarding prazosin emphasizes the need for more studies with larger samples before conclusions can be drawn.

Cyproheptadine did not show a significant therapeutic effect. Still, Morgenthaler et al. [31] highlight cyproheptadine as one of the pharmaceuticals that can be prescribed for PTSD-related nightmares. Their recommendations are, in addition, based on findings from case studies and non-placebo-controlled studies; still, Morgenthaler et al. [31] acknowledge that the literature is contradictory. Neither clonazepam nor doxazosin were associated with a significant treatment effect. It should be noted that doxazosin is very similar to prazosin in terms of mechanisms of action [64], and should therefore not yet be written off as a relevant pharmacological intervention for nightmares.

All the drug interventions in the included studies address PTSD-related nightmares. It is questionable whether the findings can be generalized beyond this study group. Comparing the effects of pharmaceutical treatment of nightmares for idiopathic and PTSD-related nightmares could be of interest. Women were strongly underrepresented in the included studies, even though women report more nightmares than men [11,26,28], are overrepresented in populations with PTSD, and report higher symptom levels in terms of anxiety and depression, both being associated with an increased incidence of nightmares [82,83]. Therefore, future studies should aim for a more even distribution of gender so that the findings can also be generalized to women with greater certainty.

Furthermore, there were only four studies [37,39,45,46] out of a total of fourteen in which the samples consisted of populations with PTSD where the PTSD-related nightmares were unrelated to war. Based on this, it is questionable whether the findings of the present meta-analysis can be generalized to populations suffering from PTSD-related nightmares in general.

As far as we know, none of the included studies were based on diagnostic criteria for nightmare disorder. It is conceivable that many of the participants in the included studies would meet these criteria, but this still makes our findings difficult to generalize to populations with nightmare disorder in general. The studies included in this meta-analysis also had other limitations. Very few of them had long-term follow-up after treatment, only two studies [41,75] were anchored in a priori power calculations, and few studies were considered to have a low risk of bias. Overall, this indicates that most of the studies had noticeable methodological limitations. The dropout rate was also relatively high in the included studies, with an average of 20% dropping out before the end of treatment.

Some limitations of the current meta-analysis should also be noted. The language restrictions may have led us to miss out on some relevant studies. Databases for gray literature were not used in the literature search and may have led to overestimating the effect [83]. On the other hand, it has been argued that unpublished data are unlikely to impact meta-analyses significantly [84,85]. It has also been argued that identifiable unpublished data are not representative of unpublished data in general [70]. The findings regarding nightmare symptomatology in the present meta-analysis were significant, and Orwin’s Fail-safe N [71] showed that 19 studies with no effect were necessary to bring the overall effect size down to a trivial level (*g* = 0.20). When using the Knapp–Hartung adjustment, the overall effect size still remained significant, but when limiting the analysis to studies with a low risk of bias, the overall effect size became non-significant. This may suggest that the findings are not very robust. Still, we argue that the overall effect size is of less importance, and recommend putting more emphasis on the effect sizes calculated for the drugs separately when interpreting the results. Despite a high degree of agreement (75.7%) between the raters in terms of risk of bias, it is also worth noting that the updated version of the Cochrane Risk of Bias 2 tool (RoB 2). [86] has been criticized for producing a low degree of inter-rater reliability, in addition to being complex and challenging to apply [87]. Most of the included studies in our meta-analysis are based on small samples. A strength of our meta-analysis is that we used a “random-effects model”. Noble [88] states that this method is preferable to a “fixed-effects model” in testing null hypotheses. Meta-analyses, in general, are further highlighted as a good tool for investigating the extent to which experimental findings can support or disprove connections and effects [87].

It should be noted that many studies with other designs than RCTs have been used to investigate treatment effects of pharmacological interventions for nightmares [89]. Still, we limited the current review to RCTs, as this design is regarded as the standard to provide unbiased estimates of treatment effect [90].

## 5. Conclusions

Even though we are starting to gain some knowledge about the treatment of nightmares, the field still lacks knowledge. When it comes to the various pharmacological therapies, prazosin is the one we have the best reasons to recommend, even though only a moderate effect of prazosin was found in the present meta-analysis. Nabilone and hydroxyzine both show promising effects, but there are too few controlled studies to conclude about their applicability. Still, these pharmaceuticals should be of interest for further research. The present meta-analysis does not provide a basis for recommending treatment with clonazepam or cyproheptadine.

For future research, trials of pharmacological interventions should be based on proper power analysis, include adverse drug effects in the analysis, aim for more even gender distributions, and include longer follow-ups. Inclusion of multiple outcomes, comprising nightmare frequency, distress, and intensity, is encouraged. Comparing the effects of pharmacological interventions on PTSD-related versus idiopathic nightmares will also be of interest. Future RCTs should also compare the effect of pharmacological and psychological interventions, separately and in combination.

## Figures and Tables

**Figure 1 ijerph-20-00777-f001:**
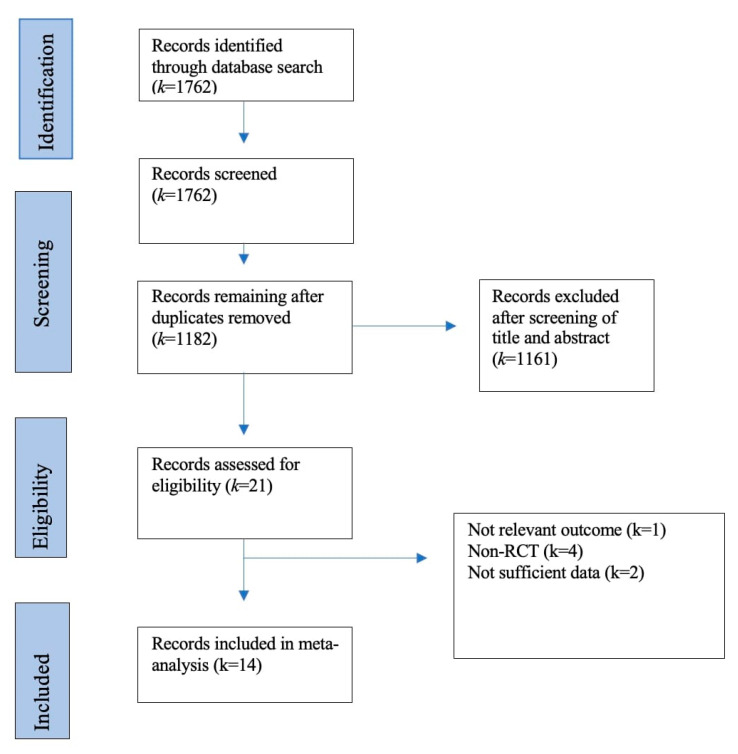
Flow chart of the different phases of the systematic literature search.

**Figure 2 ijerph-20-00777-f002:**
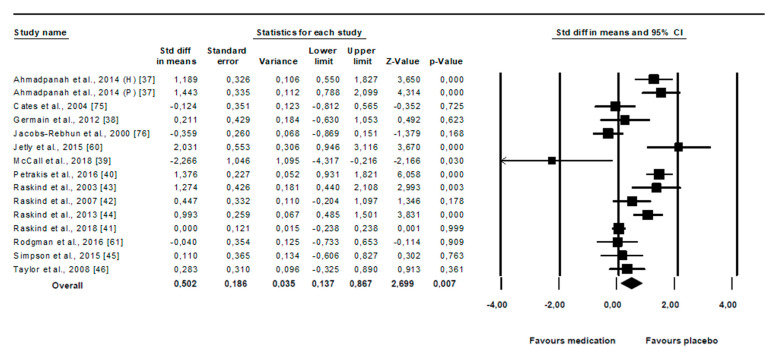
Forrest plot [37,38,39,40,41,42,43,44,45,46,60,61,75,76].

**Figure 3 ijerph-20-00777-f003:**
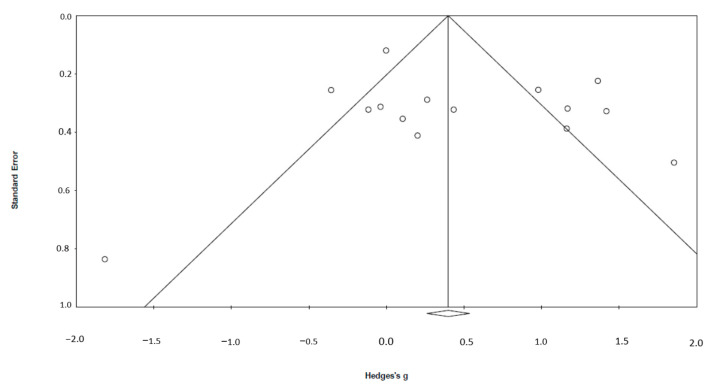
Funnel plot showing distribution of Hedges’ *g* in relation to standard error.

**Table 1 ijerph-20-00777-t001:** Study Characteristics of the Included Studies.

Author, Year, Country	Design and Duration	Population, Problem	Sample (Medication, Control Group)	Age M (SD)	Gender % Women	Dosing	Diagnostic Procedure	% Dropout	Reported Strength Calculation
Ahmadpanah et al., 2014, Switzerland [37]	Between-groups 8 weeks	Patients with PTSD. PTSD-related nightmares	Prazosin (*n* = 33) Hydroxyzine (*n* = 34) Placebo (*n* = 33)	36.18 (7.09) 36.12 (6.05) 34.21 (6.05)	24 27 33	1–15 mg 1–100 mg	PSQI ^1^	0	No
Cates et al., 2004, USA [74]	Crossover 2 weeks	Patients with war-related PTSD, PTSD-related nightmares	Clonazepam (*n* = 6) Placebo (*n* = 6)	52 52	0	1–2 mg	CAPS ^2^	0	Yes
Germain et al., 2012, USA [38]	Between-groups 8 weeks	Military veterans, chronical PTSD-related nightmares	Prazosin (*n* = 18) Placebo (*n* = 15) CBT ^6^ (*n* = 24)	39.4 (11.9) 43.6 (14.0)	11 0	1–15 mg	Sleep diary, number of recorded nightmares per week	35	No
Jacobs-Rebhun et al., 2000, USA [75]	Between-groups 2 weeks	Military veterans, chronical PTSD-related nightmares	Cyproheptadine (*n* = 34) Placebo (*n* = 35)	-	0 0	-	PSQI	13	No
Jetly et al., 2015, Canada [60]	Crossover 7 weeks	Military personnel with PTSD, PTSD-related nightmares	Nabilone (*n* = 10) Placebo (*n* = 10)	43.6 (8.2) 43.6 (8.2)	0	0.5–3 mg	CAPS	10	No
McCall et al., 2018, USA [39]	Between-groups 8 weeks	Suicidal PTSD patients, PTSD-related nightmares	Prazosin (*n* = 10) Placebo (*n* = 10)	36.3 (15.9) 43.2 (12.7)	90 80	1–25 mg, men 1–12 mg, women	DDNSI ^3^	70	No
Petrakis et al., 2016, USA [40]	Between-groups 12 weeks	Veterans with PTSD and alcoholism, PTSD-related nightmares	Prazosin (*n* = 50) Placebo (*n* = 46)	44.5 (13.2) 43.4 (12.95)	8 4.44	2–16 mg	CAPS	44	No
Raskind et al., 2003, USA [43]	Crossover 9 weeks	Military veterans, PTSD-related nightmares	Prazosin (*n* = 10) Placebo (*n* = 10)	53 (3) 53 (3)	0	1–10 mg	CAPS	20	No
Raskind et al., 2007, USA [42]	Between-groups 8 weeks	Veterans with chronical PTSD-related nightmares	Prazosin (*n* = 20) Placebo (*n* = 20)	56 (9) 56 (9)	5 5	1–15 mg	CAPS NFQ ^4^	15	No
Raskind et al., 2013, USA [44]	Between-groups 6 weeks	Soldiers and veterans, PTSD-related nightmares	Prazosin (*n* = 32) Placebo (*n* = 35)	30 (6.6) 30.8 (6.5)	19 11	1–25 mg, men 1–12 mg, women	CAPS	16	No
Raskind et al., 2018, USA [41]	Between-groups 10 weeks	Veterans, chronical PTSD-related nightmares	Prazosin (*n* = 152) Placebo (*n* = 152)	52.3 (13.8) 51.4 (13.8)	2 2	1–20 mg, men 1–12 mg women	CAPS	11	Yes
Rodgman et al., 2016, USA [61]	Crossover 16 days	Veterans, chronical PTSD-related nightmares	Doxazosin XL (*n* = 8) Placebo (*n* = 8)	34.8 (8.3) 34.8 (8.3)	0	4–16 mg	CAPS	13	No
Simpson et al., 2015, USA [45]	Between-groups 6 weeks	Patients with PTSD and alcoholism, PTSD-related nightmares	Prazosin (*n* = 15) Placebo (*n* = 15)	43.1 (11.3) 43.5 (12.4)	40 33.3	1–8 mg	PCL ^5^	33.3	No
Taylor et al., 2008, USA [46]	Crossover 3 weeks	Patients with PTSD, PTSD-related nightmares	Prazosin (*n* = 13) Placebo (*n* = 13)	49 (10) 49 (10)	85	1–6 mg	CAPS	0	No

^1^: PSQI—Nightmare symptomatology in Pittsburgh Sleep Quality Index. ^2^: CAPS—«Recurrent distressing dreams»-from the Clinician-Administered PTSD Scale. ^3^: DDNSI—Disturbing Dreams and Nightmare Severity Index. ^4^: NFQ—Nightmare Frequency Questionnaire. ^5^: PCL—PTSD Checklist-Civilian version. ^6^: CBT—Cognitive behavior therapy.

**Table 2 ijerph-20-00777-t002:** Risk of Bias in the Included Studies.

Study	Randomization	Intervention	Missing outcome data	Measurement of outcome	Reported results	Overall
Ahmadpanah et al., 2014 [37]			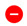			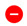
Cates et al., 2004 [75]		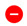				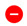
Germain et al., 2012 [38]						
Jacobs-Rebhun et al., 2000 [76]				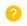	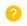	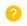
Jetly et al., 2015 [60]	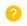					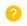
McCall et al., 2018 [39]						
Petrakis et al., 2016 [40]				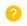		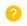
Raskind et al., 2003 [43]	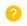					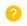
Raskind et al., 2007 [42]			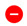			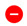
Raskind et al., 2013 [44]						
Raskind et al., 2018 [41]						
Rodgman et al., 2016 [61]						
Simpson et al., 2015 [45]	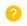		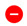			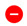
Taylor et al., 2008 [46]	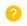	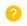				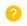

*Randomization*: biases that unfold in the randomization process. *Intervention*: biases due to deviations from intended interventions. Missing outcome data: biases due to missing outcome data. *Measurement of outcome*: biases in measurement of outcome. *Reported results*: biases in the selection process of reported results. *Overall*: skewness assessment of the study in its entirety. 

: Low risk 
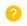
: Uncertain 
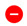
: High risk.

## Data Availability

Not applicable.

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
