# Peer review of "The Effects of Pharmacological Treatment of Nightmares: A Systematic Literature Review and Meta-Analysis of Placebo-Controlled, Randomized Clinical Trials"

_ijerph, 2022, doi:10.3390/ijerph20010777_

Round 1

Reviewer 1 Report

The authors did a literature review and meta-analysis on an interesting topic about the treatment of nightmares and concluded that under the various pharmacological therapies, prazosin is the one recommend, even though only moderate effect was found. The meta-analysis included 14 articles out of 1762 the authors found. The statistical methods used are relevant and appropriate.

This paper is well organized and written. I read this paper with great interest. I only have a few comments that hope can improve the paper.

(1)    The caption of Figure 1 is missing.

(2)    Please consider re-generating Figure 3 with a white background.  Because the black background is hard to read.

(3)    You performed the meta-analysis based on 14 articles. But the investigated drugs in these clinical trials are different (similar as heterogeneous issue), which will have different treatment effects (scale) that may lead to meaningless estimates of effects. Could you please explain how did you address or avoid this risk? 

(4)    Sensitivity analyses for the meta-analysis could be added to check the robustness of the conclusion, which also made the paper more comprehensive. One sensitivity analysis, for instance, would examine the effects of applying various meta-analysis approaches. The effects of including or removing studies from meta-analyses based on different inclusion/exclusion criteria, methodological quality, or variance may be examined in another sensitivity analysis. If the results vary between sensitivity analyses, it may be necessary to evaluate the result cautiously.

Thanks.

Reviewer 2 Report

it is quite interesting topic, however please see some comments below to improve the quality of the paper

Abstract
it is well written and covers all elements. 

however, plz include overall result from systematic review as well apart from reporting effect size result of meta analysis. so readers can have a basic understanding. 

Introduction: Please include some statistics on the nightmare burden and healthcare cost in the management of this condition. 

Please include any complementary and alternative therapies for the management of nightmares (traditional use) 

Introduction: study objective/aim in the last paragraph can be further improved. 

Methods: How the duplicate studies were removed? was it done manually or using some software? please describe in methods. 

Results: Please cite the studies in the table 

Discussion: please describe further on the possible reasons of the heterogenicity in results. 

Round 2

Reviewer 1 Report

No further comment from me.  Good luck.